# A Preparative Method for the Isolation of Calponin from Molluscan Catch Muscle

**DOI:** 10.3390/ijms23147993

**Published:** 2022-07-20

**Authors:** Stanislav S. Lazarev, Ulyana V. Shevchenko, Vyacheslav A. Dyachuk, Ilya G. Vyatchin

**Affiliations:** Laboratory of Cell Biophysics, A.V. Zhirmunsky National Scientific Center of Marine Biology, Far Eastern Branch, Russian Academy of Sciences, 17 Palchevsky Str., 690041 Vladivostok, Russia; ss_lazarev@live.ru (S.S.L.); shevchenkouv15@gmail.com (U.V.S.); slava.dyachuk@ki.se (V.A.D.)

**Keywords:** molluscan catch muscles, molluscan thin filaments, catch muscle calponin extraction, actin-activated myosin Mg^2+^-ATPase activity

## Abstract

We describe the development of a preparative method to isolate molluscan catch muscle, calponin. This method is based on the ability of calponin to interact with actin in a temperature-dependent manner. After extracting thin filaments, as previously described, the extract was ultracentrifuged at 2 °C. While other surface proteins of thin filaments co-precipitated with actin, calponin, along with some minor contaminants, remained in the supernatant. Calponin was purified through cation-exchange chromatography. The yield of pure protein was four-fold higher than that achieved through high-temperature extraction. To evaluate functionally isolated proteins, we determined the effect of calponin on Mg^2+^-ATPase activity of hybrid and non-hybrid actomyosin. The degree of ATPase inhibition was consistent with previously published data but strongly dependent on the environmental conditions and source of actin and myosin used. Furthermore, at low concentrations, calponin could induce the ATPase activity of hybrid actomyosin. This result was consistent with data indicating that calponin can modulate actin conformation to increase the relative content of “switched on” actin monomers in thin filaments. We assume that calponin obtained by the isolation method proposed herein is a fully functional protein that can both inhibit and induce the ATPase activity.

## 1. Introduction

Calponins are multifunctional proteins involved in modulating the function of actin in smooth muscle and non-muscle cells [1]. All calponins are thermostable, and are capable of interacting with actin in the fibrillar form (F-actin) [2,3] and inhibiting the Mg^2+^-ATPase activity of actomyosin regardless of Ca^2+^ concentration [4]. Three calponin isoforms have been identified in vertebrate tissues (h1, h2, and h3), one of which is specific for differentiated smooth muscle cells (h1) [5,6]. The role of the h1 calponin isoform is assumed to be closely related to the latch phenomenon: the state where muscle maintains a steady tension for a very long period of time with a low energy cost [7,8,9].

The molecular mechanism underlying latch contraction remains unclear [10]; however, it has been suggested that calponin is involved in the tethering of thick- and thin-filaments, like molluscan twitching [11]. The latch state of vertebrate smooth muscles is similar to the catch state of molluscan smooth muscles [11], which also contain calponin-like proteins [12,13]. Matrix Assisted Laser Desorption/Ionization/Mass spectrometry/mass spectrometry (MALDI/MSMS) data indicate that bivalve calponin is homologous to h1 and binds to antibodies against this calponin [13]. The physicochemical properties of these proteins are also similar [12,13,14]. Bivalve calponin was shown to markedly inhibit the activity of actomyosin Mg^2+^-ATPase, as well as vertebrate calponin [12,13,14]. However, the mechanism underlying this inhibition is different, with molluscan calponin being a competitive inhibitor of actoS1 ATPase [15]. The function of bivalve calponin was investigated by polarization fluorometry in ghost fibers isolated from skeletal muscle [14,15,16]. Calponin can influence the ATPase cycle by changing the actin conformation. Calponin may be involved in the regulation of catch muscle contraction in molluscs [10].

Studies on the molecular genetics of catch muscle calponin have revealed the full length complementary DNA sequence from *Mizuhopecten yessoensis* [17], *Crenomytilus grayanus* [18], and the pearl oyster *Pinctada fucata* [10]. Studies have investigated the tissue expression of these proteins, including the distribution and number of isoforms [10,19]. Isoforms of calponin at 34 and 40 kDa have been identified in the catch muscle of the mussel *C. grayanus* [13].

Despite extensive studies on molluscan smooth muscle calponin, a preparative method for isolation of this protein has been proposed for the first time. The alternative methods described in the literature [12] either entail significant protein losses during isolation (see Discussion) or are an adaptation of the heat-extraction method developed for vertebrate smooth muscles [13] with a relatively low yield of calponin (not published). Here, we propose an alternative isolation method, which does not subject the resulting protein to harsh treatment and generates a high yield. All stages of this method have been shown to be reproducible.

## 2. Results

### 2.1. Rigorization and Washing of Mussel Muscles 

The rigorization stage is the first and most important step in isolation of proteins from mussel thin filaments. With this stage is not performed, a significant portion of thin filaments is lost [20]. Here, all procedures were performed on ice. A 100 g sample of fresh minced adductor mussel from *C. grayanus* was rigorized for 24 h with four volumes of glycerol solution (50% glycerol, 50 mM NaCl, 2 mM MgCl_2_, 2 mM egtazic acid [EGTA], 3 mM NaN_3_, 0.1 mg/mL trypsin inhibitor, 0.5 mM phenylmethylsulphonyl fluoride [PMSF], 1 mM DTT, and 20 mM Imidazole-HCl, pH 6.5) with continuous agitation on an overhead stirrer. Upon completion, glycerinated muscles were diluted up to 1 L with washing solution containing 50 mM KCl, 1 mM MgCl_2_, 0.2 mM EGTA, 1 mM NaN_3_, 1.0 mM DTT, 2.5 μg/mL leupeptin (LPN), and 10 mM phosphate buffer, pH 6.5. The suspension was ground in a meat grinder and homogenized on a Polytron PT 2500E homogenizer (Kinematica, Malters, Switzerland) at 4800 rpm for 10 min. The homogenate was diluted up to 4 L with washing solution and centrifuged at 14,000× *g* for 20 min. The resulting pellet was resuspended in 1 L of the washing solution and homogenized using a Polytron PT 2500E (4800 rpm for 3 min). The homogenate was recentrifuged as described above. The pellet was used as rigorized “myofibrils”.

### 2.2. Mussel Calponin Fraction

The last washing pellet was suspended in 450 mL of extraction solution, containing 75 mM KCL, 5 mM MgCl_2_, 5 mM EGTA, 15 mM ATP, 5 mM sodium pyrophosphate, 1 mM DTT, 0.1 mM PMSF, 2.5 μg/mL LPN, and 10 mM imidazole-HCl, pH 7.0. The extraction was performed for 40 min with stirring, and the extract was obtained by centrifugation on an Allegra X-30R centrifuge (Beckman, Brea, CA, USA) at 10,864× *g* for 30 min (Beckman rotor, type F0685). The collected extract (about 400 mL) was clarified at 28,305× *g* for 30 min (Beckman rotor, type F0630) to remove contaminating thick filament proteins. The clarified extract included thin filaments of catch muscle and contained about 800 mg of protein (Figure 1A, lane 1). To obtain calponin, the thin-filament extract was centrifuged on a Beckman Optima L-90K at 110,000× *g* for 120 min, (Beckman rotor, type 50.2Ti). During centrifugation, the temperature was set around 2 °C to prevent actin and calponin from interacting. 

This procedure resulted in precipitation of the natural actin–tropomyosin–troponin complex (Figure 1A, lane 3), while all the catch muscle calponin was obtained in the supernatant (Figure 1A, lane 2). As shown by electrophoresis, the calponin fraction also contained around 450 mg of other proteins. Some proteins were removed using the isoelectric precipitation procedure, keeping the samples on ice until the pH reached 4.75. Following incubation for 30 min, the aggregated protein was clarified at 13,975× *g* for 20 min. Clarified calponin was extracted by salting with solid ammonium sulfate of 0–33% saturation and collected with centrifugation at 13,975× *g* for 30 min. The 0–33% fraction was dissolved in 20 mL of column buffer (30 mM KCl, 0.2 mM EGTA, 2 mM NaN_3_, 20 mM Imidazole-HCL, 0.5 mM DTT, and 6 M urea, pH 6.50) and dialyzed against three changes of 25 volumes of this solution. The dialyzed fraction was clarified by centrifugation on Allegra X-30R (Beckman, Brea, CA, USA) at 10,864× *g* for 20 min (Beckman rotor, type F0685). The final step of calponin purification was chromatography.

### 2.3. Calponin Chromatography

The dialyzed fraction (270 mg) was applied to a CM Sephadex C-50 (Sigma-Aldrich, St. Louis, MI, USA) packed column (1 × 25 cm), equilibrated with column buffer. Calponin was absorbed on resin, while other contaminating proteins passed through (Figure 1B). The column was eluted with a linear KCl gradient from 0.03 to 0.5 M KCl. Both calponin zones (40 and 34 kDa) were eluted in a single peak beginning around 180 mM KCl (Figure 1B,C). The fractions containing calponin were day-long dialyzed against the standard or high-salt dialyze solution (75 or 500 mM KCl, 2 mM MgCl2, 0.2 mM EGTA, 2 mM NaN3, 0.5 mM DTT, and 25 mM imidazole-HCl, pH 6.5).

The chromatographically pure calponin yield at this stage was about 1.25 mg per 1 g of muscle, which is equivalent to 125 mg of protein in a volume of 30 mL. Therefore, calponin can be lyophilized in the presence of 10% sucrose.

### 2.4. Calponin Solubility

Despite the high efficiency of the calponin isolation method described above, use of the obtained protein was complicated by low solubility in solutions with low ionic strength. As shown in Figure 2, the soluble/insoluble protein ratio was less than 50% at a concentration of 0.25 mg/mL, which is close to that used in reconstruction experiments of contractile models. When concentrations reached a level convenient for protein storage (3–4 mg/mL), calponin solubility dropped to 12%. To overcome this, it was suggested to increase the ionic strength of the calponin preparation up to 500 mM KCl; this approach is commonly used for the storage of proteins polymerized at low ionic strength (myosin, myorod). Under these conditions, calponin solubility exceeded 80%. Notably, heat treatment also seemed to increase calponin solubility. Following 40 s heat treatment of chromatographically pure calponin with low solubility (4 mg/mL in presence of 75 mM KCl), solubility increased from 11 to 65% (Figure 2, the last column). Heat treatment was applied by placing a small volume (3–4 mL) of the material in boiling water for 40 s.

### 2.5. Properties of Mussel Calponin

The ability of calponin to inhibit the ATPase activity of actomyosin was used to evaluate the functionality of calponin obtained by the above method [12,13,15]. Experience obtained investigating ATPase inhibition by tropomyosin of the catch muscle was used to evaluate the effect of calponin on the actin–myosin interaction [21]. In that case, an inhibitory effect of mussel tropomyosin was found to be an artifact of the hybrid contractile model. Nevertheless, comparison of protein behavior between hybrid and non-hybrid models has contributed to our understanding of the underlying mechanism. Therefore, calponin has been tested in contractile models of different hybrids (Figure 3).

In the contractile model “native” to calponin (Figure 3A, mussel myosin + mussel actin), calponin presented the lowest degree of ATPase inhibition, at about 30%. The maximum degree of inhibition was observed in a contractile model reconstructed using vertebrate myosin and molluscan actin (79%). The contractile models reconstructed using the skeletal muscle actin presented intermediate results (32% with molluscan myosin and 73% with skeletal muscle myosin).

As shown above, ionic strength has a marked effect on calponin solubility. Therefore, to clarify whether decreasing ionic strength to values below 75 mM KCl would prevent the calponin–actin interaction, the inhibition experiment was repeated with 30 mM KCl (Figure 3B). Surprisingly, under these conditions, inhibition of ATPase activity in the contractile model containing molluscan myosin did not decrease (as would be expected with a decline in calponin solubility); in contrast, this increased by 6–20% depending on the actin used. Furthermore, the results obtained with actomyosin reconstructed using skeletal muscle myosin were ambiguous, with inhibition increasing from 79 to 84%, while with skeletal muscle, actin inhibition decreased by 10%. Notably, in those cases, the degree of ATPase inhibition by calponin was higher in the contractile models reconstructed using molluscan actin. This phenomenon is explained by the fact that mussel calponin binds to rabbit actin much weaker than to mussel actin (Figure 4).

## 3. Discussion

The calponin isolation method described in this study was a continuation of a previously reported method for obtaining globular actin from molluscan catch muscles [22].The previous method was modified by changing a stage of high ionic strength, during which all surface proteins are removed from thin filaments at once, to a two-stage “undressing” of actin (low and high ionic strength). This approach is based on the fact that calponin does not interact with actin at low temperatures [13]. Therefore, ultracentrifugation of the thin filament extract at 2 °C in a medium with low ionic strength would cause this protein to remain in the supernatant, being contaminated slightly (Figure 1A, lane 2). In this case, most thin filament proteins would precipitate (Figure 1A, lane 3). The subsequent procedure of calponin purification exploits the basic properties of the protein.

A major advantage of the presented method is the opportunity it provides to continue the procedure to obtain other thin-filament proteins from the precipitated complex. In this case, pellets containing the complex should be suspended in a small volume of extraction solution supplemented by KCl to 600 mM, followed by 2 h centrifugation at 110,000× *g* on a Beckman Optima L-90K (Beckman, USA) (Beckman rotor, type 70Ti). The supernatant obtained via this centrifugation method contains the troponin–tropomyosin complex, which is separated as described earlier [20]; pellets contain pure actin, which can be depolymerized [22].

The efficiency of the presented isolation method is high. While calponin production by heat-extraction [13] provides a yield of about 0.34 mg/g (not published), the method described above provides a yield almost four-fold higher. Unfortunately, it is not possible to compare the amount of calponin we obtained with that produced by a method developed for muscles of another bivalve mollusc, *Mytilus galloprovincialis* [12]. Nevertheless, we assumed that protein losses during the production process are significant. There are two explanations for this. First, the lack of a glycerinization stage [22], which results in the loss of a sizable amount of thin filaments [13,23]. Second, the approach includes dialysis of muscle extract against a solution with a low ionic strength (30 mM KCl) aimed to remove salt-soluble proteins. Due to the solubility of calponin under conditions of low ionic strength, this step can lead to its aggregation and loss.

The solubility of the resulting calponin deserves special attention. We did not experience the low solubility of mussel calponin reported during calponin isolation with heat treatment [13]. Figure 2 may explain the underlying reason for this. Nevertheless, the factors that caused calponin solubility to increase upon heating remain unclear, and may include modification of the protein by heating. Therefore, we recommend avoiding heat-treatment during calponin isolation. In this regard, the approach proposed in our study seems to be optimal, since it excludes heat treatment of the obtained protein.

A remaining issue is how calponin is used in the reconstructed models.

During development of the contractile model, the ionic strength decreased from the high values at which calponin (500 mM KCl) is stored, to the target level of 75 mM or even 30 mM KCl. For this reason, there was concern regarding the partial transition of calponin into an insoluble form. This could result in a marked decrease or loss of calponin’s ability to inhibit the actomyosin interaction. However, in most models, calponin demonstrated a significant level of overall inhibition (Figure 3). In some models, calponin inhibited ATPase activity markedly at a calponin-to-actin ratio of 1:2, while inhibition was close to maximal at a ratio of 1:1. These findings indicate good solubility and functionality of calponin in the models.

Notably, the degree of inhibition in the “native” contractile model at 30 mM KCL is similar to that for calponin (CaP) obtained by an alternative method from muscles of *M. galloprovincialis*: 43% and 41% of inhibition with a 2:1 CaP:actin ratio [12]. Moreover, this relationship is consistent with that obtained for molluscan calponin isolated by heat extraction [13].

Although the major goal of elucidating the effect of isolated calponin on ATPase activity was to confirm its functionality, a more in-depth comparison of the results obtained in hybrid models is of interest. A general conclusion that can be derived from all considered models is that the degree of maximum ATPase inhibition by calponin is determined primarily by the source of myosin, and not actin (a difference of 41–49% vs. 2–6% at 75 mM and 25–34% vs. 12–21% at 30 mM KCL). The shape of the inhibition curves is determined by the type of actin used. Thus, in many cases, differences between the inhibition values in the contractile models using the molluscan and vertebrate actin are significant, especially when low and medium amounts of calponin are added. Thus, in most cases, despite the same calponin content, the ATPase activity of the mussel actin-containing model can be inhibited to an almost maximum extent, while the model with vertebrate actin can be inhibited to a rather low extent and can even be activated. Since the difference in inhibition rates decreases significantly with increasing calponin content, the observed differences can be explained by an almost two-fold difference in the affinity of calponin for mussel and rabbit actins (Figure 4).

The existence of a stage at which calponin increases ATPase activity in the vertebrate contractile model is unexpected. However, it is even more surprising that, with the KCl concentration varying from 75 to 30 mM KCl, the same stage appears in the contractile model reconstructed using molluscan actin. It is doubtful that a qualitative change in the ATPase activity curve was caused by a decrease in calponin solubility. We observed the stage at which calponin enhances the actin–myosin interaction rather than a stage of protein accumulation (as observed for models reconstructed using molluscan myosin). Thus, these results can be interpreted on the basis of the data reported by Borovikov et al., who observed similar effects when studying the conformational changes in actin produced by calponin [16]. In the hybrid contractile model, under conditions with a low calponin content (1 CaP:3 actin) and a low ionic strength (10 mM KCl), they showed calponin to be capable of modulating the actin–myosin interaction through inhibition at the stage of strong binding and facilitation at the stage of weak binding. Moreover, in the presence of ATP, calponin can increase the relative amount of incorporated actin monomers, which increases the relative number of myosin heads capable of forming strong binds with actin. Therefore, it is likely that the driving force for the change in inhibition curves in the presence of 75 to 30 mM KCl is the change in protein affinity; in particular, a change in the balance whereby calponin influences actin switching between two conformational states (on and off).

Thus, calponin obtained as a result of the proposed isolation method is able to inhibit the ATPase activity of actomyosin. Moreover, under some conditions, this protein can induce ATPase activity. These data are due to the hybridity of the contractile model, which demonstrates that such models can bring unpredictable results.

## 4. Materials and Methods

### 4.1. Protein Isolation

Proteins were isolated from the posterior adductor of the sea mussel *Crenomytilus grayanus* and muscles of the rabbit hind limbs and back. Mussel F-actin (actin isolated from muscle in the fibrillar form), was prepared as described previously [22]. Catch muscle myosin was isolated and prepared as reported earlier [24]. Rabbit skeletal muscle myosin was prepared as previously described [25]. The acetone powder of the rabbit skeletal muscles was prepared as described earlier [22]; actin was extracted as described previously [26]. Rabbit skeletal muscle tissues were provided by the G.B. Elyakov Pacific Institute of Bioorganic Chemistry, FEB RAS (Vladivostok, Russia). Protein purity was controlled by electrophoresis; neither myosin nor actin preparations contained visible contaminants. Protein concentration was determined by the micro-biuret method [27].

### 4.2. Mg^2+^-ATPase Assay

Myosin and actin with (or without) calponin (not clarified) were mixed in solution with a high ionic strength (500 mM KCl, 2 mM MgCl_2_, 0.1 mM CaCl_2_, 0.5 mM DTT, 2 mM NaN_3_, and 25 mM imidazole-HCl, pH 6.5). Then, the ionic strength was reduced to 30 or 75 mM KCl by adding a solution with the same composition, but without KCl. The reconstruction was completed by incubating the suspension for 10 min at +4 °C and 10 min at 25 °C. The reaction was initiated by adding Mg^2+^-ATP to 0.3 mM and terminated after 10 min by adding trichloroacetic acid to 5 mM. Inorganic phosphate was determined colorimetrically [28]. Data reproducibility was determined by conducting the experiment in triplicate. When plotting the graphs, the data were averaged using GraphPadPrism (San Diego, CA, USA) with associated confidence intervals.

### 4.3. Protein Binding

Chromatographically pure and clarified actin and calponin were mixed as described for the Mg^2+^-ATPase assay (calponin concentration was 0.1 mg/mL and lower). Following incubation at 25 °C for 10 min, proteins were ultracentrifuged at the same temperature (at 110,000× *g*, for 120 min, on a Beckman Optima L-90K ultracentrifuge (USA); rotor type 70Ti). Pellets were separated by SDS-PAGE as described previously [13]. The gels were stained with Coomassie blue R250. Relative contents of actin and calponin were determined as described previously [20]. Images were obtained by scanning the wet gels using an office scanner. The scans were processed in the TotalLab TL120 program (UK). Color intensity of the calponin and actin zones was determined without calculating the protein dye binding coefficients. According to low calponin solubility, which can affect co-precipitation results, control values gained after sole calponin sedimentation were subtracted from the data. Based on the results, relationships between the amount of calponin bound to actin and the known amount of calponin added were identified. The procedure was repeated at least three times for three different gels for each protein; the average result is shown in Figure 4.

### 4.4. Calponin Solubility

Solubility of the calponin preparation was determined after sedimentation of the insoluble fractions on an Eppendorf MiniSpin centrifuge (12,100× *g*, 20 min, 2 °C), Germany. The soluble and insoluble fractions of calponin were compared by determining the distribution of calponin between the supernatant and the pellet. The protein content was determined by the micro-biuret method [24]. As a control, the data were compared with the known initial calponin content in the preparation.

## Figures and Tables

**Figure 1 ijms-23-07993-f001:**
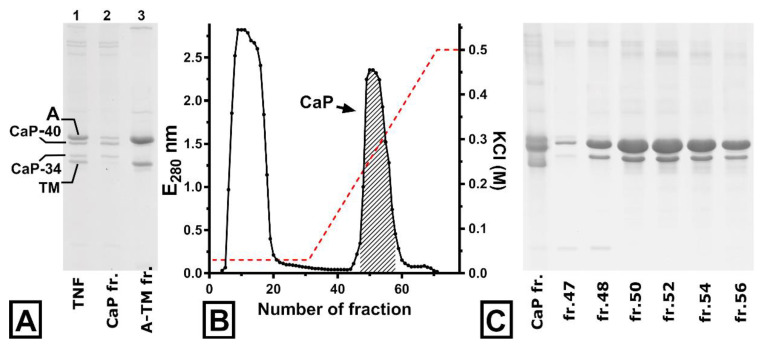
Sodium dodecyl-sulfate polyacrylamide gel electrophoresis (SDS–PAGE) analysis of preparations obtained through isolation of calponin from mussel muscle: (**A**), fractionation of thin-filament extract (lane 1) on the calponin-containing (lane 2) and actin–tropomyosin (lane 3) fractions; (**B**), purification of molluscan calponin through ion-exchange chromatography; Red line shows ionic strength changing during elution. (**C**), electropherogram showing the results of calponin chromatography. A, actin; TM, tropomyosin; CaP, calponin.

**Figure 2 ijms-23-07993-f002:**
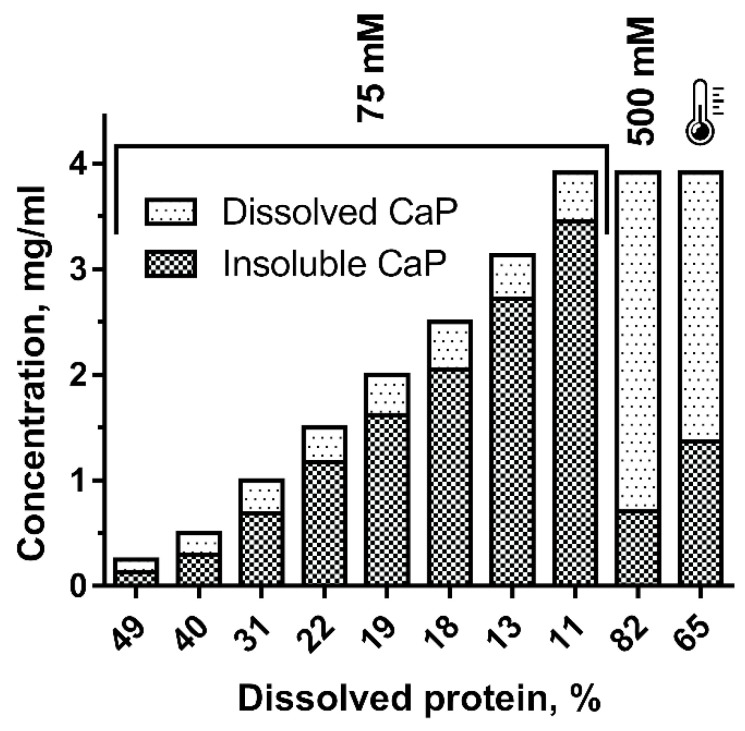
Relationship between calponin solubility and protein concentration in the presence of 75 or 500 mM KCl and after heating. Ionic conditions are indicated at the top of the histogram. The thermometer indicates the results obtained for calponin heat-treatment.

**Figure 3 ijms-23-07993-f003:**
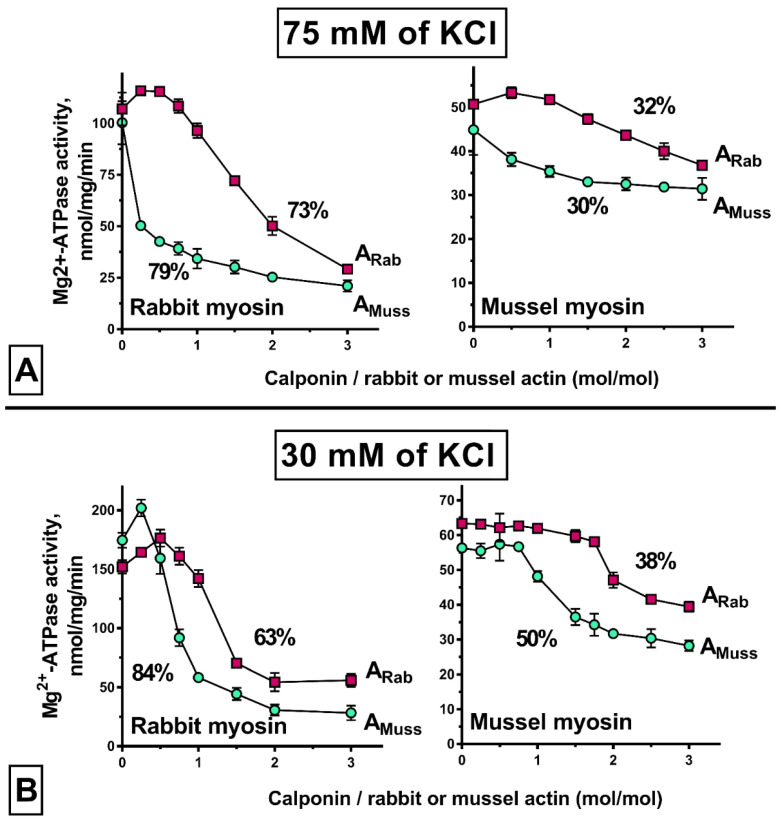
Effect of molluscan calponin on the ATPase activity of actomyosin reconstructed using vertebrate (rabbit) and molluscan (mussel) proteins in the presence of 75 mM (**A**) and 30 mM (**B**) KCl. Concentrations are as follows: mussel or rabbit myosin, 0.1 mg/mL; mussel or rabbit F-actin, 0.1 mg/mL. A_Rab_, actin of rabbit; A_Muss_, actin of mussel.

**Figure 4 ijms-23-07993-f004:**
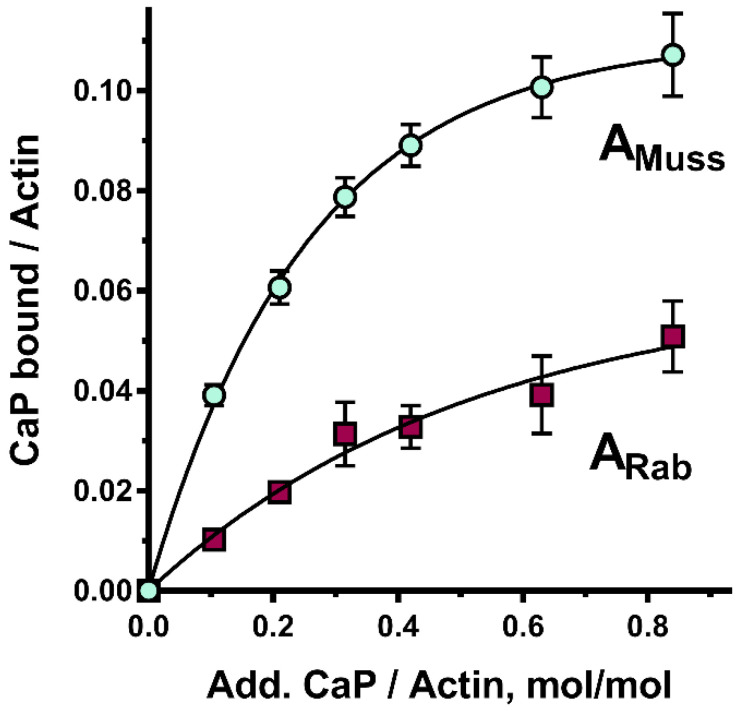
Co-precipitation of molluscan calponin with molluscan (A_Muss_) and vertebrate (A_Rab_) actin.

## Data Availability

Not applicable.

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
