# Peer review of "A Preparative Method for the Isolation of Calponin from Molluscan Catch Muscle"

_ijms, 2022, doi:10.3390/ijms23147993_

Round 1

Reviewer 1 Report

This is a very straightforward paper describing how calponin can be extracted form  molluscan catch muscle in high yield. The methodology is clearly described but the identification of the product is inadequate.  The functional significance of calponin in molluscan catch muscle needs to be more fully discussed. 

Introduction

1               Can the authors expand on the relationship of calponin to latch and catch.  The references provided are very old and somewhat speculative- has any hard evidence been produced in recent years?

Results

1               The authors need to determine the quantity of calponin relative to actin, tropomyosin and troponin in the thin filaments starting material.  Is the native quantity sufficient to have functional effects?

2               The calponin product needs to be positively identified by mass spectrometry. In particular you should determine if the 34k band is a degradation product of the 40k band or a separate isoform.  The Genebank sequences of these putative proteins should be quoted and matched to the MS.

3               All the experiments use mixed 40k and 34k calponin.  In the  solubility experiment does the ratio remain constant or is one form less soluble than the other.  If 34k is a degradation product it would be quite likely to be more prone to aggregation than the native protein.

4               The experiments showing calponin as an inhibitor need large quantities of calponin (> 1 per actin).  Is there enough calponin in molluscan muscle to have any significant effect?  Figure 4 suggests calponin does not bind strongly to actin; what about actin-tropomyosin-troponin? 

Note that an inhibitory function in vertebrate smooth muscle was ruled out on this basis (Marston, (1991) 10.1016/0014-5793(91)80862-w)

Discussion

1               Calponin is not a well known contractile protein and calponin in molluscs is still more a niche subject , therefore more background is need for an effective discussion.  Please describe:

-       The molluscan calponin genes and patterns of expression. How they are related to vertebrate calponin?

-       Location of calponin in the molluscan cell and the quantities expressed?

-       What is known about its function in molluscs relative to vertebrates, where its involvement in cell signalling seems to be more important than its effect on contractile proteins?

Author Response

Question

Can the authors expand on the relationship of calponin to latch and catch. The references provided are very old and somewhat speculative- has any hard evidence been produced in recent years?

Response

Unfortunately, we were unable to identify any solid evidence linking calponin and catch. Based on recent literature, the most relevant articles we identified were references 14, 15, 16, and 18. The first three articles clearly demonstrate the ability of calponin to influence actomyosin interaction by changing the conformational structure of actin (14, 15, 16). The last Reference 18 indicates the identification of the binding site of calponin with twitchin, a protein of the catch muscle, very likely having the most direct relation to catch. Thus, we do not doubt the involvement of calponin in the functioning of the contractile compartment of the catch muscle; however, its exact purpose remains unclear.

Results

Question

The authors need to determine the quantity of calponin relative to actin, tropomyosin and troponin in the thin filaments starting material.  Is the native quantity sufficient to have functional effects?

Response

We have previously studied the relationship between the thin filament proteins in troponin identification of the catch muscle. Unfortunately, due to a lack of separation of the two calponin isoforms, the reliability of the data is limited and has not been reported. Considering the binding coefficients of the electrophoretic dye by both the 34 and 40 kDa isoforms of calponin, its ratio with respect to actin is about 1 : 3. In this content, given the presence of two actin-binding sites in calponin (Mino et al., 1998), it seems quite likely that this protein exerts functional effects. However, when the procedure for separating the two isoforms of calponin is debugged, we will perform densitometry of fine threads for calponin.

Question

The calponin product needs to be positively identified by mass spectrometry. In particular you should determine if the 34k band is a degradation product of the 40k band or a separate isoform.  The Genebank sequences of these putative proteins should be quoted and matched to the MS.

Response

Both isoforms of calponin have been identified using mass spectrometry (reference 13) and the isoform sequence has been uploaded to GenBank (reference 18); work on isoform sequence 34 is currently underway. In addition to differences in the sequence of these calponins, the two isoforms are evidenced by the constancy of the calponin preparation and the stability of the ratio between isoforms, which does not change among samples, especially during storage.

Question

All the experiments use mixed 40k and 34k calponin.  In the  solubility experiment does the ratio remain constant or is one form less soluble than the other.  If 34k is a degradation product it would be quite likely to be more prone to aggregation than the native protein.

Response

The ratio between the isoforms of calponin, as I indicated above, remains stable in the preparations. This also applies to solubility experiments. We were interested in the possibility of differing solubility of calponin 34 and 40 kDa, as it could provide a basis for their separation. However, it turned out not to be so.

Question

The experiments showing calponin as an inhibitor need large quantities of calponin (> 1 per actin).  Is there enough calponin in molluscan muscle to have any significant effect?  Figure 4 suggests calponin does not bind strongly to actin; what about actin-tropomyosin-troponin? Note that an inhibitory function in vertebrate smooth muscle was ruled out on this basis (Marston, (1991) 10.1016/0014-5793(91)80862-w)

Response

Regarding the inhibitory activity of calponin, the data presented herein, as well as data presented in the literature, lead us to question whether the inhibitory effect of calponin on ATPase is the main function of this in the contractile apparatus. Furthermore, as noted above, the role of this protein has not yet been clarified; consequently, the effect of calponin on ATPase activity may be a side effect of the processes activated by it in the contractile apparatus. Therefore, we used the inhibitory effect to indicate the compliance of calponin with those obtained using other approaches. To achieve some qualitative effects (activation of ATPase by 15% or inhibition by 50%, see Figure 3A, left), the use of calponin with a native ratio to actin (3A: 1CaP) may be sufficient.

Figure 4 shows the difference in the binding of calponin with different actins, but not the binding of the troponin-tropomyosin complex with actin. This type of interaction was not an objective of this study. Nevertheless, we have studied the binding of troponin components with actin and found that the limiting factor for saturation was the reduced troponin content in the contractile apparatus and not the weak binding between proteins.

Question

Discussion

Calponin is not a well known contractile protein and calponin in molluscs is still more a niche subject , therefore more background is need for an effective discussion.  Please describe:

The molluscan calponin genes and patterns of expression. How they are related to vertebrate calponin?

Location of calponin in the molluscan cell and the quantities expressed?

What is known about its function in molluscs relative to vertebrates, where its involvement in cell signalling seems to be more important than its effect on contractile proteins?

Response

The topics proposed for discussion, particularly data on calponin genes in comparison with similar proteins of other muscles, would be more relevant for an article investigating the function of calponin and its possible role in the catch. Therefore, the consideration of these topics in this paper seems redundant and would distract from our work and the results obtained. However, we will consider these suggestions for the discussion in a review on proteins of the locking muscle, which is currently in development.

Reviewer 2 Report

This paper reports the new preparation method for molluscan calponin from the catch muscle. The reviewer thinks it is worth being published, but there are some concerns to be addressed as follows. The reviewer would like the authors to revise the manuscript before its publication.

1.       L104-107 The column buffer used for chromatography contained 6M urea. The reviewer would like to know the reason why 6M urea was added.

2.       L115-121 The reviewer would like to know if the calponin denatured by 6M urea was able to be refolded by the dialysis against the standard or high-salt dialyze solution. If so, how much calponin was refolded? All of the calponin was successfully refolded?

3.       L127 What does ‘solubility’ mean in this paper? Does the soluble calponin mean fully refolded protein after the dialysis above? Is the insoluble calponin denatured protein? Or does the calponin purified in this study behave like a salt-soluble protein such as myosin or paramyosin, which is insoluble in the low salt buffer while it is native?

4.       L139 When was heat treatment carried out?

5.       L308 The reviewer would like to see the control data of the co-sedimentation assay. When only calponin was centrifuged, no calponin had been precipitated? If calponin got insoluble during the incubation for 10 min, it would be precipitated. This possibility should be denied.

Author Response

Question

L104-107 The column buffer used for chromatography contained 6M urea. The reviewer would like to know the reason why 6M urea was added.

Response

The addition of urea at such a concentration is necessary to separate calponin from impurities during ion exchange chromatography. A study on the separation of proteins at a lower concentration revealed unsatisfactory results; therefore, it was decided to stop at a concentration of 6 M.

Question

L115-121 The reviewer would like to know if the calponin denatured by 6M urea was able to be refolded by the dialysis against the standard or high-salt dialyze solution. If so, how much calponin was refolded? All of the calponin was successfully refolded?

Response

A comparison of calponin dialyzed for urea removal in solutions with low and high ionic strength revealed that the percentage of soluble calponin was increased under higher ionic strength compared with lower ionic strength (82 vs. 11%). Nevertheless, further dialysis of a protein soluble in high ionic strength in a solution with an ionic strength of 75 mM KCl decreased its solubility to 11% (data not shown).

These data indicate that low solubility is a property of calponin in the absence of impurities, rather than due to disruption of its structure following urea removal. Nevertheless, we have not investigated protein structure after chromatography. However, the structural integrity of the protein can be determined by functional testing of calponin for the ability to inhibit myosin ATPase. Thus, the effects of the resulting calponin do not differ compared to protein not subjected to chromatography, but rather obtained by thermal extraction.

Question

L127 What does ‘solubility’ mean in this paper? Does the soluble calponin mean fully refolded protein after the dialysis above? Is the insoluble calponin denatured protein? Or does the calponin purified in this study behave like a salt-soluble protein such as myosin or paramyosin, which is insoluble in the low salt buffer while it is native?

Response

Here, solubility is a measure of the transition of protein to sediment during centrifugation at low speeds (12000× g). The use of non-soluble proteins (apart from suspicions of its denaturation) is difficult in experiments; difficulties arise with the accurate and correct selection of aggregated material. With sufficiently rigid centrifugation, it is difficult to resuspend insoluble protein, even in the presence of solvent agents (urea, KCl). Indeed, it is most likely that calponin behaves more like a salt-soluble protein similar to myosin and paramyosin. This is evidenced by the transition of calponin to an insoluble form, as described above, following the removal of salt.

Question

L139 When was heat treatment carried out?

Response

The heat treatments were performed using calponin already subjected to purified chromatographic, with a low intrinsic solubility of 11%. We have made adjustments to the text to make this clearer.

Question

L308 The reviewer would like to see the control data of the co-sedimentation assay. When only calponin was centrifuged, no calponin had been precipitated? If calponin got insoluble during the incubation for 10 min, it would be precipitated. This possibility should be denied.

Response

Yes, indeed, some part of calponin, when it was precipitated under the described conditions, remained insoluble. These data were considered as a control of calponin deposition in the absence of actin. However, the percentage of such insoluble protein in the control was very low. This was due to a combination of two factors: the protein used was already soluble due to the presence of high ionic strength and the concentration of calponin in the experiment did not exceed 0.1 mg/mL. At these values, the solubility of calponin is quite high (at least 50%). We thank the reviewer for this comment and have included some discussion of this in the appropriate section.

Round 2

Reviewer 1 Report

The authors have addressed all the criticisms adequately.  Given the limited objectives of the manuscript it is good piece of work.